# Synergistic photoredox and copper catalysis by diode-like coordination polymer with twisted and polar copper–dye conjugation

Yusheng Shi[1,3], Tiexin Zhang [1,3], Xiao-Ming Jiang[2], Gang Xu [2], Cheng He[1] & Chunying Duan [1✉]

Synergistic photoredox and copper catalysis confers new synthetic possibilities in the pharmaceutical field, but is seriously affected by the consumptive fluorescence quenching of Cu (II). By decorating bulky auxiliaries into a photoreductive triphenylamine-based ligand to twist the conjugation between the triphenylamine-based ligand and the polar Cu(II)–carboxylate node in the coordination polymer, we report a heterogeneous approach to directly confront this inherent problem. The twisted and polar Cu(II)–dye conjunction endows the coordination polymer with diode-like photoelectronic behaviours, which hampers the inter- and intramolecular photoinduced electron transfer from the triphenylamine-moiety to the Cu(II) site and permits reversed-directional ground-state electronic conductivity, rectifying the productive loop circuit for synergising photoredox and copper catalysis in pharmaceutically valuable decarboxylative C(sp$^3$)–heteroatom couplings. The well-retained Cu(II) sites during photo-irradiation exhibit unique inner-spheric modulation effects, which endow the couplings with adaptability to different types of nucleophiles and radical precursors under concise reaction conditions, and distinguish the multi-olefinic moieties of biointeresting steride derivatives in their late-stage trifluoromethylation-chloration difunctionalisation.

[1] State Key Laboratory of Fine Chemicals, Dalian University of Technology, Dalian 116024, China. [2] State Key Laboratory of Structural Chemistry, Fujian Institute of Research on the Structure of Matter, Chinese Academy of Sciences, Fuzhou 350002, China. [3]These authors contributed equally: Yusheng Shi, Tiexin Zhang. ✉email: cyduan@dlut.edu.cn

C(sp³)–heteroatom bond formation is of fundamental importance in pharmaceutical fields, however, remains one of the major challenges in the fields of cross-coupling chemistry[1]. Taking advantage of the multiple valance states, Cu (II) catalysts enable the capture of single-electronic carbon-centred radicals, and the following facile reductive elimination to realise the key steps of cross couplings[2]. The synergy of photoredox and copper catalysis provides a sustainable way to generate radicals for conducting C(sp³)–heteroatom couplings[3,4]. The major challenging limits the direct combination of photoredox and copper catalysis was the strong quenching effect of Cu(II) ions towards the highly reductive excited states of photosensitizers that required for the radical generation[5]. In practice, photoredox and copper catalysis are usually compromised by adopting the high loadings of Cu(I) precursors to kinetically balanced off the excited-state quenching effect of in situ generated Cu(II) ions[6–8]. Thinking outside the box by a manner of aggregation state, the crystalline porous coordination polymers[9,10] fix photosensitizers and copper(II) ions in high local concentrations and spatially isolate them to block the futile intermolecular fluorescence quenching (Fig. 1b). Moreover, the electronic communications between chromophore-based ligands and Cu(II) nodes could be regulated to hamper the intramolecular fluorescence quenching. The well-modified Cu(II)–dye coordination polymers would directly confront the inherent requirements of this regime in a heterogeneous manner and simultaneously circumvent the risk of residual heavy metals that caused by high loadings of Cu(I) salts.

The pioneering results revealed that the connection modes between photosensitizers and redox-active metals in coordination supramolecular systems[11,12] vitally affect the electron transfer routes between both entities. The direct connection by a conjugative linkage permits the molecular wire-like bidirectional electronic conductivity at both excited and ground states[13], which impairs its catalytic applications. Twisting the conjugative junction between the electron-donating and accepting sections within a single-molecule device was shown to markedly enhance the charge-transfer resistance[14,15]. Noted that higher reorganisation energy was required for one-way electron transfer through the high-polar carboxylate–metal node in the most common carboxylate-based coordination polymers[16]. Moreover, as a versatile photosensitiser, triphenylamine (TPA) has been successfully modified into carboxylate-based coordination polymers to generate radicals by photoreduction[17]. Thus, the twisted conjugative connection of TPA-based ligand and polar carboxylate–copper node might realise the diode-like unidirectional electronic conductivity within coordination polymer for compromising photoredox and copper catalysis[18]. Herein, we show a new approach to modifying the electronic communications in the Cu(II)–TPA coordination polymer by introducing a bulky group at the ortho-position of the phenylcarboxylic coordination group of TPA-based ligand to twist the π-conjugation between phenyl moiety and the coordinated carboxylate (Fig. 1a). We envision that this special series connection of twisted conjugation and polar carboxylate–copper node within Cu(II)–TPA coordination polymer should kinetically alleviate the inter- and intramolecular photoinduced electron transfer (PET) from the excited-state TPA fragment to Cu(II) ion[19]. The ground-state electronic communication thermodynamically allows the alternative directional single electron transfer (SET) from the in situ formed Cu(I) to the oxidised TPA moiety (Fig. 1c)[20]. This molecular diode-like behaviour of twisted conjugation and polar node rectifies the productive loop-circuit electron transfer route that required by synergising the photoredox and copper catalytic cycles (Fig. 1c).

Considering the value of decarboxylative C(sp³)–heteroatom couplings in the pharmaceutical field[21] and their reliance on both photocatalyitc radical generation and Cu(II)-catalytic radical capture as well as the following reductive elimination, the decarboxylative C(sp³)–N couplings between iodonium carboxylate-type alkyl radical precursors and N-centred heterocycles are examined as a proof-of-concept of the diode-like Cu(II)–TPA coordination polymer for synergising photoredox and copper catalysis (Fig. 1d). Taking advantages of Cu(II) centres that retained during the PET processes to exert the possible activation to the ester moiety, this synergistic catalysis is further extended to the use of more challenging N-hydroxyphthalimide esters as alkyl radical precursors for the value-added C(sp³)–N/O/S couplings with aniline/phenol/thiophenol nucleophiles, respectively. Moreover, this unique activation of the enriched ground-state Cu(II) sites is further unveiled to successfully distinguish the multiple olefinic moieties in the late-stage trifluoromethylation-chloration difunctionalisation of olefins within biointeresting steride derivatives[22,23].

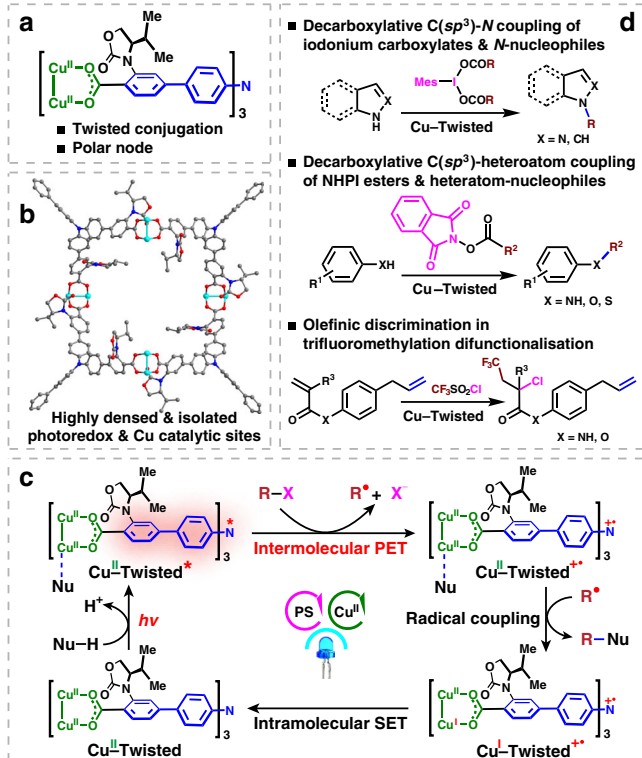

**Fig. 1 Diode-like Cu(II)–dye rectifier for synergistic photoredox and copper catalysis.** Schematic illustrations of **a** the steric-induced twisted conjugation between polar Cu(II)–carboxylate node and photoreductive **TPA**-moiety and **b** the densed fixation and spatial isolation of photoredox and Cu(II) sites in coordination polymer. **c** The mechanistic perspective of synergistic catalysis by Cu(II)–dye, and **d** the heterogeneous catalytic applications.

## Results

**Synthesis and characterisation of coordination polymers.** The installation of a chiral oxazolidinone moiety[24] at ortho-position of carboxylic linker of the typical **TPA**-based ligand tris(4′-carboxybiphenyl)amine (named as H₃L–Planar) afforded a new ligand tris[4-(4-carboxy-3-((R)-4-isopropyl-2-oxooxazolidin-3-yl)-phenyl)phenyl]amine (named as H₃L–Twisted). The coordination polymer Cu–Twisted was obtained in a 70% yield by the solvothermal reaction between Cu(NO₃)₂·3H₂O and H₃L–Twisted at 80 °C for 3 days (Supplementary Data 1). Cu–Twisted

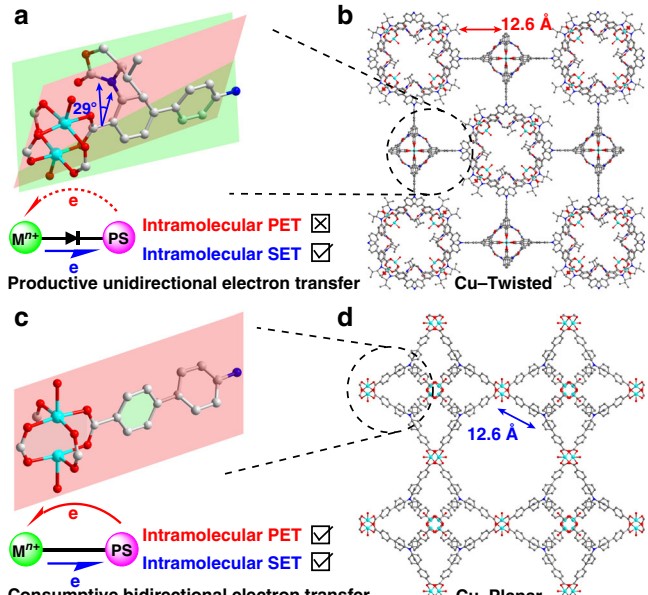

**Fig. 2 Coordination polymers with/without twisted conjugation. a** The twisted dihedral angle and **c** the coplanar conjugation between Cu (II)-carboxylate coordination plane and neighbouring phenyl plane of Cu–**Twisted** and Cu–**Planar**, respectively. The insets exhibited the diode-like (for **a**) and molecular wire-like (for **c**) electron transfer routes. The 3D porous networks of **b** Cu–**Twisted** and **d** Cu–**Planar**. Cu cyan, C grey, O red, N blue, H atoms were omitted for clarity.

consisted of truncated cubic cages (internal diameter ca. 2.9 nm, Supplementary Fig. 2), each delimited by 8 **L-Twist**$^{3-}$ linkers and 12 $Cu_2(O_2C)_4$ paddle-wheels, forming dumbbell-shaped 1D channels with a cross-sectional area of $15 \times 27$ Å$^2$ along the $a$-direction (Fig. 2b)[25]. The free volume of desolvated Cu–**Twisted** was estimated as ca. 62% of the porous polymer, and a Brilliant Blue R-250 dye uptake experiment with Cu–**Twisted** yielded a 54% absorption amount of the coordination polymer weight as determined by UV–vis spectroscopy (Supplementary Fig. 9), implying the possibility of accommodating substrates and reagents within the pores[26]. The steric hindrance between the bulky oxazolidinone and the $Cu_2(O_2C)_4$ paddle-wheel forged a twisted dihedral angle between the conjugated carboxylic coordination group and the adjacent phenyl moiety (Fig. 2a)[27], which induced the symmetry of coordination polymer into a pto lattice (Fig. 2b and Supplementary Fig. 2)[28]. This distortion together with the polar carboxylate–copper node provided the possibility to finely modify the electronic communication for the kinetic alleviation of fluorescence quenching of Cu(II) towards the highly reductive excited-state **TPA** moiety[18].

In the absence of bulky auxiliary, a control catalyst was prepared by solvothermal reaction between $Cu(NO_3)_2 \cdot 3H_2O$ and $H_3$**L-Planar** (45% yield). The (3,4)-connected three dimensional (3D) networks (internal diameter ca. 3.8 nm, Supplementary Fig. 4) were assembled between the tritopic ligands and 4-connected $Cu_2(O_2C)_4$ paddle-wheels in a high-symmetry tbo topology (Supplementary Table 1 and Supplementary Data 2), similar to the well-known HKUST-1[29]. Notably, the absence of axial rotational restriction from the bulky auxiliary allowed the conformation of coplanar conjugation between the polar Cu (II)–carboxylate node and the adjacent phenyl moiety of ligand (Fig. 2c). Thus, this control coordination polymer was named as Cu–**Planar**. It should be noted that the existence or not of distortion along the axial direction of ligand did not affect the identical Cu–N$_{TPA}$ distances (ca. 12.6 Å) in both Cu–**Twisted** and

Cu–**Planar** (Fig. 2b, d), and the non-interpenetrated nature of two coordination polymers precluded the intermolecular luminescence quenching of Cu(II) nodes to the excited state of indirectly connected **TPA**-moieties. Moreover, the vast internal cavities and open windows of both Cu(II)–**TPA** coordination polymers were believed to facilitate the rapid mass transfer during their catalytic applications.

Electrochemical analyses of both Cu–**Twisted** and Cu–**Planar** exhibited the peaks at ca. 1.0 V corresponding to the redox potentials of the **TPA**-based ligands (Supplementary Fig. 11). The similar reductive potentials of the excited-state frameworks were determined as −1.49 V and −1.59 V for Cu–**Twisted** and Cu–**Planar**, respectively (Supplementary Table 2), based on the free energy change ($E^0$) between the ground and vibrationally relaxed excited states (Supplementary Fig. 12)[30], demonstrating that both the excited-state coordination polymers can theoretically reduce iodomesitylene dicyclohexanecarboxylate (abbreviated as MesI (OCOCy)$_2$, $E_{1/2}^{red} = -1.14$ V vs. SCE), one of the prominent iodonium carboxylate-type radical precursors, for the generation of alkyl radicals[7].

Electrochemical impedance spectroscopy (EIS) of the two coordination polymers exhibited different arc radii in the high-frequency regions, demonstrating that the charge-transfer resistance ($R_{ct}$) of Cu–**Twisted** (ca. 4.8 kΩ) was roughly three times larger than that of Cu–**Planar** (ca. 1.6 kΩ) (Fig. 3a). The photocurrent responses of two coordination polymers suggested their well reproducible photocurrents upon on/off cycles of the light irradiation (Fig. 3b). Compared with Cu–**Twisted**, a significant enhancement in the photocurrent response was observed for the case of Cu–**Planar**, indicating more efficient intramolecular PET process. Clearly, the twisted conjunction mode did not affect the thermodynamic photoreducing abilities of coordination polymers, but remarkably altered the kinetic features of intramolecular electronic communications.

The UV–vis absorption spectra of solid-state Cu–**Twisted** and Cu–**Planar** exhibited the bands covering a broad visible range (400–550 nm, Supplementary Fig. 10) that attributed to the absorption of **TPA**-based ligands, and the typical d–d transition peaks of dicopper paddle-wheel units were observed above 600 nm. Upon photoirradiating, Cu–**Twisted** exhibited weak intramolecular quenching of Cu(II) nodes towards the highly reductive excited-state **TPA** moieties (Supplementary Fig. 15). The remarkably lower fluorescence intensity and shorter fluorescence lifetime (2.1 ns vs. 2.9 ns, see Fig. 3c) of Cu–**Planar** than those of Cu–**Twisted** under the identical conditions suggested the more efficient intramolecular fluorescence quenching in the coordination polymer Cu–**Planar**.

Electron paramagnetic resonance (EPR) studies revealed that both coordination polymers exhibited the characteristic signals of Cu(II) ions with $g$ value of ca. 2.09 (Fig. 3d)[31]. Photoirradiation with compact fluorescent lamps for 15 min lead to significant depression of Cu(II) signal in EPR spectra of Cu–**Planar**. In comparison, the identical photoirradiation did not cause obvious Cu(II) signal variation in EPR spectra of Cu–**Twisted**, confirming that the twisted conjugation between **TPA**-moiety and polar carboxylate–copper node effectively hampered the intramolecular PET process between them (Fig. 2a, c). Moreover, the much greater amount of Cu(II) sites that retained in the photoirradiation of diode-like Cu–**Twisted** was believed to benefit the Cu(II) catalytic steps in its combination with photocatalysis (Fig. 1c).

**Heterogeneous decarboxylative C(sp$^3$)–heteroatom coupling.** Noted that the carboxylic acids are more abundant, stable, and less toxic chemical feedstocks compared with alkyl halides, the classical alkyl radical precursor, thus the redox-activated

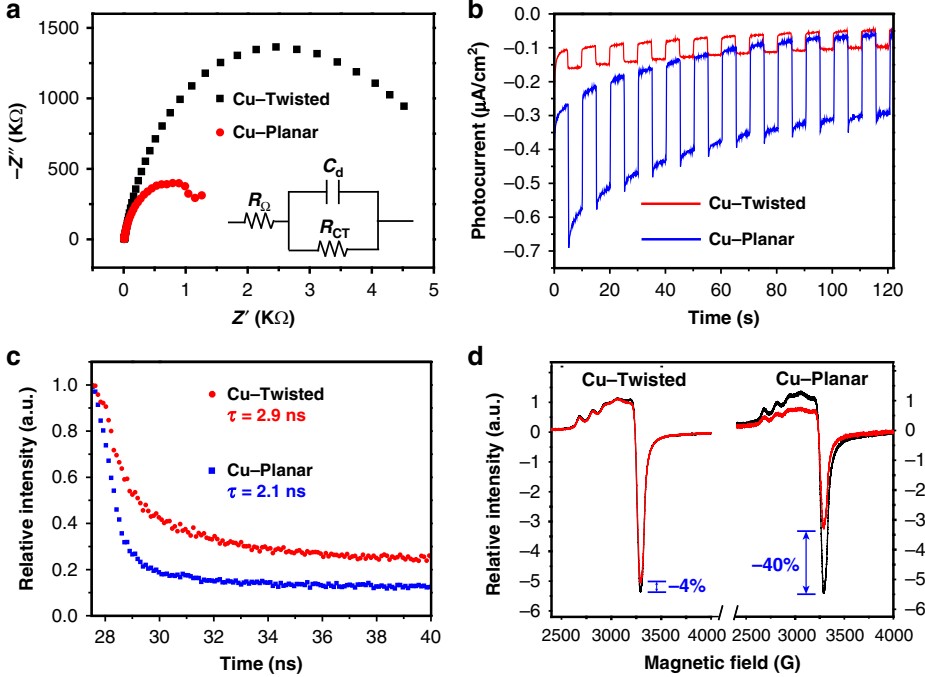

**Fig. 3 Characterisation of the diode-like photoelectronic behaviours.** Comparisons of **a** EIS, **b** transient photocurrent responses, **c** fluorescence lifetimes, **d** EPR spectra of Cu–**Twisted** and Cu–**Planar**, respectively.

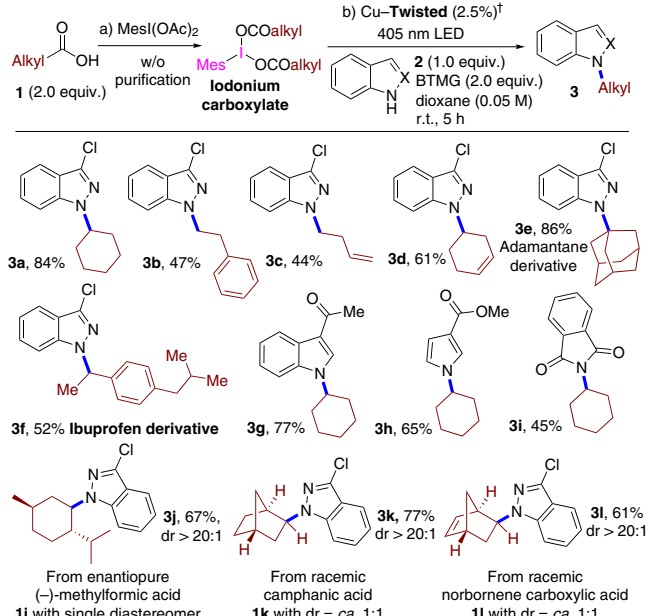

**Fig. 4 Decarboxylative C(sp³)-N coupling by Cu-Twisted.** †Conditions: iodonium carboxylate prepared from **1** (2.0 equiv.), *N*-nucleophile **2** (0.3 mmol, 1.0 equiv.), Cu-**Twisted** (2.5%), BTMG (2.0 equiv.), dioxane (0.05 M), 405 nm LED, N₂, room temperature (r.t.), 5 h. Isolated yields.

carboxylic acids like iodonium carboxylates are considered as the practical and sustainable radical sources in this case. We next examined the synergistic photoredox and copper catalytic performances of the obtained Cu(II)–**TPA** coordination polymers using decarboxylative C(sp³)–heteroatom coupling of iodonium carboxylates and heteroatom nucleophiles as the benchmark[7]. Upon addition of MesI(OCOCy)₂ into the suspensions of Cu–**Twisted** or Cu–**Planar** in 1,4-dioxane, the luminescences of

coordination polymers were markedly quenched (Supplementary Fig. 16). This result suggested a productive intermolecular PET from the excited state of **TPA** moiety to MesI(OCOCy)₂ was allowed for the generation of alkyl radicals. In a typical procedure, a mixture of MesI(OCOCy)₂, nitrogen-centred nucleophile 3-chloroindazole **2a**, basic additive BTMG (BTMG = 2-*tert*-butyl-1,1,3,3-tetramethylguanidine), and 2.5 mol% Cu–**Twisted** in 1,4-dioxane was subjected to visible-light irradiation from a 405-nm LED under a N₂ atmosphere, and the desired C(sp³)–N coupling product was obtained in 84% isolated yield after 5 h (Figs. 4 and 3a). Subsequently, a series of iodonium carboxylates derived from the primary linear, secondary acyclic and cyclic, and tertiary substituted alkyl carboxylic acids were found applicable in this protocol to deliver the corresponding *N*-alkyl heteroaryl products **3** in medium to good efficiencies (Figs. 4 and 3a–f). Especially, the use of tertiary carboxylic acid **3e** successfully introduced a sterically demanding adamantine moiety into the product in 86% yield; it should be noted that *N*-alkylation employing tertiary alkyl halides is elusive using traditional nucleophilic substitution. Interestingly, an ibuprofen derivative valuable to medicinal chemistry was accessible by this methodology (**3f**). In addition to indazole, other nitrogen heterocycles like indole, pyrrole, and phthalimide were also amenable to this *N*-alkylation protocol (**3g**–**3i**). This catalytic system well tolerated functional groups such as terminal (**3c**) or internal (**3d**) olefins, which were inherently susceptible to radical addition, implying that Cu–**Twisted** might harness the behaviour of radical species in an inner-spheric manner.

Control experiments demonstrated that coordination polymer catalyst and light irradiation were indispensable for the coupling reaction (Supplementary Table 3, entries 2 and 3), neither the dark condition nor the absence of Cu–**Twisted** gave the formation of **3a**. There was no noticeable further conversion after hot filtration of Cu–**Twisted**, suggesting the heterogeneous nature of reaction (Supplementary Table 3, entry 14). After photocatalysis, the coordination polymer was easily isolated from

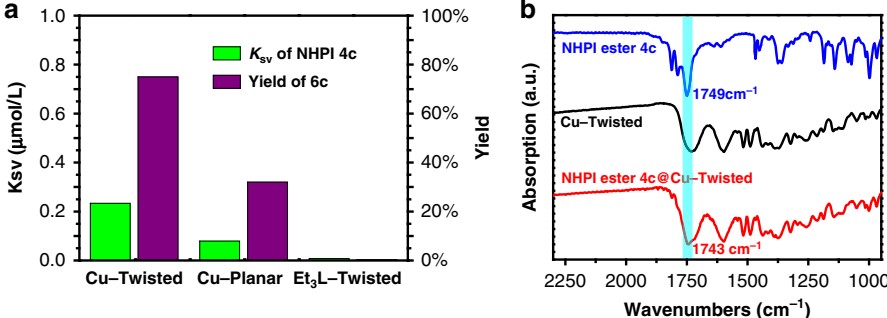

**Fig. 5 Coordination activations of Cu(II) sites in Cu(II)–TPA. a** Comparison of fluorescence quenching (upon addition of NHPI ester **4c**) and catalytic efficiencies of Cu–**Twisted**, Cu–**Planar**, and ester form of ligand. **b** Comparative IR studies on coordination activations of framework. The ester stretching vibrations of free and adsorbed **4c** were highlighted by the inserted turquoise bands.

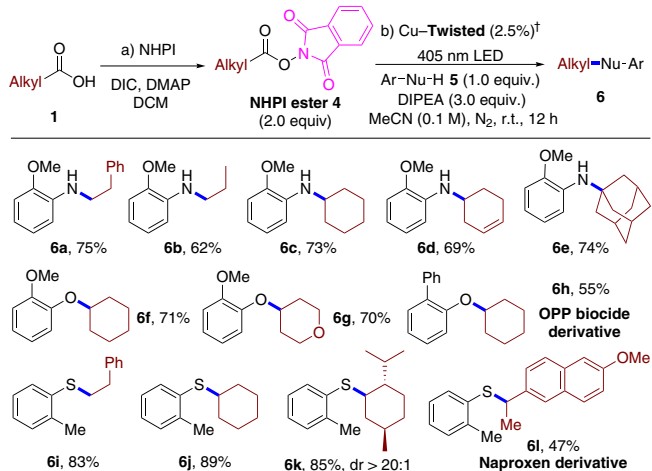

**Fig. 6 Decarboxylative C(sp³)–heteroatom coupling by Cu–Twisted.**
†Conditions: Cu–**Twisted** (2.5%), **5** (0.3 mmol, 1.0 equiv.), **4** (2.0 equiv.), DIPEA (3.0 equiv), MeCN (0.1 M), 405 nm LED, N₂, r.t., 12 h. Isolated yields.

the reaction mixture by centrifugation and could be reused at least three times without a marked decrease in reactivity (83–79%; Supplementary Fig. 26). The PXRD pattern of the recovered catalyst indicated that it maintained its structural integrity (Supplementary Fig. 22). The nearly similar catalytic performances of the pristine crystals of coordination polymers compared to the finely ground samples ruled out the vital influence of particle sizes of heterogeneous catalysts towards catalytic efficacy (Supplementary Fig. 23), implying that the reaction mainly occurred in the pores of coordination polymer (Supplementary Table 3, entries 1, 4, 11 and 12).

When a radical scavenger tetramethylpiperidine-N-oxyl was added to the reaction mixture, the conversion was inhibited immediately (Supplementary Table 3, entry 13), suggesting a radical mechanism. The only use of ester form of ligand Et₃**L**–**Twisted** or Cu(II) salt as catalyst gave no reactions (Supplementary Table 3, entries 7 and 9), reflecting the necessity of two kinds of catalytic sites. The simple combinations of Cu(II) salt either with Et₃**L**–**Twisted** or Me₃**L**–**Planar** afforded the low conversions, probably owing to the futile intermolecular fluorescence quenching of Cu(II) salt (Supplementary Table 3, entries 5 and 6). The reference coordination polymer Cu–**Planar** gave a 41% yield of **3a**. Although this conversion was only half the level of Cu–**Twisted** (Supplementary Table 3, entries 1 and 4), at least the incorporation of **TPA** moieties together with polar Cu(II)–carboxylate nodes into coordination polymer partially

hampered the intermolecular fluorescence quenching that limited the direct combination of photoreductive dye and Cu(II) catalyst in the solution. Furthermore, if the similar redox potentials (Supplementary Table 2), substrate encapsulating abilities, and intra-pore substrate diffusion kinetics (Supplementary Fig. 25) of Cu–**Twisted** and Cu–**Planar** were also taken into account, the distinct catalytic efficiencies of two coordination polymers (Supplementary Fig. 24) might be attributed to their remarkably different intramolecular electronic communications.

The circular dichroism spectra of Cu–**Twisted** exhibited a negative signal at approximately 346 nm and positive dichroic band centred at 275 nm similar to that of H₃**L**–**Twisted** (Supplementary Fig. 8), and the Cu(II) ions here were enveloped in the semi-open chiral pores (Supplementary Fig. 2), which possibly provided stereoinduction in the radical capture of Cu(II) sites. When employing the iodonium carboxylate prepared from chiral natural derivative (1R,3R,4S)-p-menthane-3-carboxylic acid, the net resulting stereochemistry of the carboxylic α-carbon was fortunately maintained during C(sp³)–N coupling (dr. > 20:1, Figs. 4 and 3j) in the presence of proximal chiral centres on the menthane scaffold. After photoreductive decarboxylation, the frequent flipping of the formed C(sp³)-centred radical erases the original stereoinformation[32], thus it should be practical to use the radical precursors derived from the racemic α-substituted carboxylic acids. For **3k** and **3l**, chiralities were successfully introduced into the achiral α-positions of the carboxylic moieties by employing cheap racemic camphanic acid (**1k**) and norbornene carboxylic acid (**1l**) instead of their expensive enantioenriched homologues, demonstrating the inner-spheric redox process between the in situ generated radical species and the well-retained Cu(II)[33] within asymmetric local environment for controlling this diastereoselectivity.

Beside the iodonium carboxylates, the redox-active N-hydroxyphthalimide (NHPI) esters[34] derived from alkyl carboxylic acids were also the prominent alkyl radical precursors in decarboxylative cross-coupling reactions, as the surrogates for alkyl halides. Like that of iodonium carboxylate MesI(OCOCy)₂, the addition of NHPI ester **4c** into the Cu–**Twisted** suspension strongly quenched the luminescence (Fig. 5a and Supplementary Fig. 17), whereas NHPI ester **4c** hardly quenched the fluorescence of Et₃**L**–**Twisted** (with a quite low quenching constant $K_{sv} < 0.01$ μmol⁻¹ L). As the excited-state reductive potential of Et₃**L**–**Twisted** was more negative than that of Cu–**Twisted** (Supplementary Table 2), the Cu(II) sites of coordination polymer Cu–**Twisted** were speculated to improve the fluorescence quenching process of NHPI ester. IR spectra of Cu–**Twisted** crystals that soaked in the solution of NHPI ester **4c** exhibited the red-shifted carbonyl stretching peak of **4c** compared with that of the free molecule (Fig. 5b and Supplementary

Fig. 19), this potential interaction between the carbonyl of radical precursor **4c** and the Cu(II) site of coordination polymer might draw the radical precursor closer to the nearby photocatalytic centre[35], which might be correlated to the above-mentioned enhancement of fluorescence quenching effect. Similarly, a bathochromic shift in the N–H stretching vibration of *ortho*-anisidine **5a** (3460–3444 cm$^{-1}$) was also observed after substrate incubation within Cu–**Twisted**, verifying that the nitrogen atom of N–H was the nucleophilic binding site and could be activated during substrate adsorption within the coordination polymer (Supplementary Fig. 20).

With minor changes of reaction conditions, this heterogeneous synergistic catalytic system was also applicable to the coupling of diverse primary, secondary, and tertiary alkyl redox-active NHPI esters with anilines in good yields (Fig. 6a–e)[36,37]. Then, the substrate scope of this heterogeneous approach could be easily expanded from anilines to chalcogen-centred nucleophiles[38]. Phenols were successfully employed in C(sp$^3$)–O coupling (**6f**–**6h**), and the *ortho*-phenylphenol fungicide derivative **6h** was obtained in 55% yield. Further extension of chalcogen-type nucleophiles to thiophenols facilitated rapid access to thioether-derived Naproxen **6l**, showcasing the efficient late-stage functionalisation for drugs and the perfect resistance to the potential sulphur-poisoning effect.

As shown in Fig. 7, our heterogeneous synergistic catalytic strategy using a twisted-conjugated Cu(II)–dye coordination polymer supplies a unified mechanistic paradigm to meet the diversified needs of reaction kinetics between photoredox and copper catalytic cycles[3] when employing the different types of substrates and radical precursors. Upon light irradiation, the diode-like coordination polymer Cu–**Twisted** hampers the intramolecular fluorescence quenching to switch on the intermolecular PET from the excited-state **TPA** moiety of ligand to the alkyl radical precursor that possibly activated by the Cu(II) site. The in situ generated alkyl radical is subjected to an inner-spheric redox process with the Cu(II) site that coordinates with the heteroatom-centred nucleophile, delivering Cu(I) species and the desired coupling product, as depicted by the pioneering homogeneous protocols[39]. Then, the radical cation of the oxidised

**TPA** moiety retrieves one electron from the Cu(I) site through the twisted connection at the ground state, which completes the productive closed-circuit electron transfer route for the synergy of photocatalytic and copper catalytic cycles and simultaneously regenerates the resting-state Cu–**Twisted** for the next round of reaction[14]. Within the confined environments of Cu–**Twisted**, the well-retained Cu(II) sites enriches substrates and reagents to improve the local concentration of activated nucleophiles and in situ generated alkyl radicals. This key kinetic modulation is believed to furnish the heterogeneous synergistic photoredox and copper catalysis in a concise and easy handling manner, which alleviates the reliance on the varied and elaborative reaction conditions obtained by the massive screening efforts that typically needed in the homogeneous protocols when using different types of nucleophiles and radical precursors. In a comparison, the low concentration of the transient Cu(II) species that in situ generated from the homogeneous Cu(I)–photocatalyst system might hamper them from exerting the readily accessible modulation effects[3,40].

**Heterogeneous trifluoromethylation-chloration of olefins.** As mentioned above, the well-retained Cu(II) centres during PET process enabled the inner-spheric capture of single-electronic radical intermediates and the fixation of double-electronic heteroatom-centred nucleophiles and carbonyl-containing radical precursors, which shed light on different-typed value-added catalytic application by integrating nucleophiles, carbonyl-containing substrates, and radical species in a step-economic manner, such as the pharmaceutically important trifluoromethylation-chloration difunctionalisation[41] of olefins of α,β-unsaturated compounds. Appending CF$_3$ and Cl moieties to olefins was highly dependent on the copper-mediated inner-spheric processes[42,43], since that the olefinic moieties were prone to either the intermolecular radical oligomerisation/polymerisation or the intramolecular cyclisation when encountered with free radicals. The negative enough reductive potential of excited-state Cu–**Twisted** allowed the photoreduction of triflyl chloride (TfCl) ($E_{1/2}^{red} = -0.18$ V vs. SCE)[44], the typical CF$_3$ radical precursor. As a consequence, the quenching of 520-nm luminescence of the acetonitrile suspension of Cu–**Twisted** crystals upon the addition of TfCl was indicative of the PET process from excited-state **TPA**-based ligand to TfCl for the generation of CF$_3$ radicals under light irradiation (Supplementary Fig. 18).

Given the propensity of the methacrylate ester to undergo the radical oligomerisation/polymerisation[45], the phenyl methacrylate (Figs. 8a and 7a) was chosen as the model substrate of difunctionalisation by Cu–**Twisted**. In a typical procedure, a mixture of phenyl methacrylate **7a**, TfCl, basic additive (2,4,6-collidine), and 2.5 mol% Cu–**Twisted** in acetonitrile was subjected to visible-light irradiation from two household compact fluorescent lamps (CFLs), obtaining the trifluoromethylation-chloration product **8a** in an isolated yield of 92% (Fig. 8a), and no radical oligomerisation/polymerisation products could be detected. The only use of Et$_3$**L**–**Twisted** resulted in the dominative oligomerisation/polymerisation of the substrate (Supplementary Table 5, entry 7), and the simple combination of Cu(II) salt and Et$_3$**L**–**Twisted** gave a hugely diminished conversion (13%), reflecting the necessity of the *holo* coordination polymer for hampering the undesirable intermolecular fluorescence quenching (Supplementary Table 5, entry 9). When using the controlled catalyst Cu–**Planar**, a lower yield of 45% was detected, implying the importance of well-rectified intramolecular electronic communication in the diode-like catalyst for the efficient diffunctionalisation (Supplementary Table 5, entry 4). Furthermore, various α,β-unsaturated esters (**7b**–**7e**) were

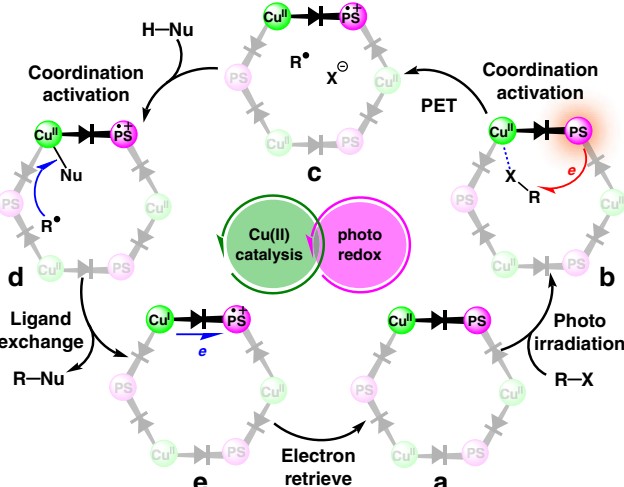

**Fig. 7 Proposed mechanism of synergistic photoredox and copper catalysis by Cu-Twisted.** The schematic illustrations of **a** initial state of Cu-**Twisted**, **b** PET process from **TPA**-moiety to the radical precursor that activated by Cu(II) site, **c** in situ generated alkyl radical within framework, **d** inner-spheric redox process between the alkyl radical and the nucleophile that coordinated to Cu(II) site, and **e** retrieval of electron from Cu(I) to radical cation form of **TPA**-moiety.

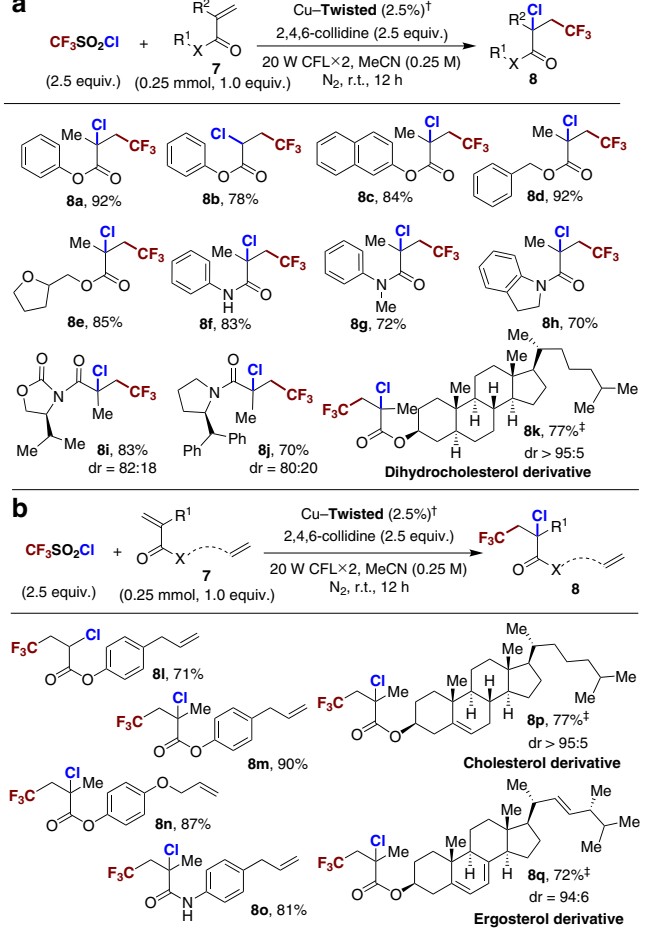

**Fig. 8 Trifluoromethylation-chloration difunctionalisation of olefins.**
**a** Substrate scope of α, β-unsaturated esters and amides. **b** Discrimination of multiple olefinic sites in difunctionalisation. †Conditions: **7** (0.25 mmol, 1.0 equiv.), TfCl and collidine (2.5 equiv.), Cu–**Twisted** (2.5%), MeCN (0.25 M), two 20-W CFLs, N₂ atmosphere, r.t., 12 h. Isolated yields. ‡1,2-Dichloroethane as the solvent.

transformed to the corresponding α-chloro-β-trifluoromethyl ester products in good to high yields. Besides, the α,β-unsaturated amides (**7f–7h**) also participated well in this difunctionalisation, and the well-known radical cyclisations with intramolecular *N*-phenyl moieties were effectively depressed (Supplementary Fig. 29)[46]. Those results indicated that the carbon-centred radical intermediates formed after CF₃ radical addition might be intercepted by the Cu(II) nodes to prohibit the side-pathways like oligomerisation/polymerisation or cyclisation. Attaching bulky auxiliaries to the neighbouring positions of carbonyl groups of the substrates should benefit the mutual stereo-recognition and induction between substrates and local environments (**7i–7k**). In particular, the reaction of **7k** bearing a bio-relevant dihydrocholesterol fragment, afforded the targeted product nearly as sole diastereomer (Fig. 8a, k).

Structurally complex small molecules containing repeating functional groups[47], like multiple olefinic natural products, have an extraordinary capacity for a wide range of useful functions[48]. The similar thermodynamic reactivities of different olefinic sites of natural derivatives made the discrimination of olefinic groups a challenging task in the presence of highly active CF₃ radicals, like in the case of trifluoromethylation-chloration difunctionalisation (Supplementary Table 6)[49]. Here, the natural flavour derivative **7l** containing a carbonyl-adjacent olefin and a carbonyl-free olefin

in the molecular scaffold was chosen as the model substrate[50]. DFT calculation revealed the thermodynamic feasibility of docking the carbonyl site of substrate **7l** through Cu–O interaction (Supplementary Table 7), with a free energy change of ca. 12.44 kcal mol⁻¹ in **7l** over the adsorption upon Cu(II) node. The IR spectra of Cu–**Twisted** with encapsulated **7l** suggested that the C=O stretching peak of **7l** was red-shifted relative to that of the free molecule (1744–1732 cm⁻¹, Supplementary Fig. 21), verifying the possible fixation of the carbonyl moiety of substrate on the Cu(II) site. As a consequence, the distances between the multiple olefinic sites of the docked substrate and the catalytic centres should be effectively differentiated within Cu–**Twisted**, which might provide an ideal model of the Hammond postulate-typed site-selectivity control (Supplementary Fig. 31)[51].

Under the typical reaction conditions of Fig. 8a, the trifluoromethylation-chloration difunctionalisation of **7l** solely occurred on the carbonyl-adjacent olefinic site and afforded the formation of product **8l** in an isolated yield of 71% (Fig. 8b), and the carbonyl-free olefinic site of allylbenzene terminal of **7l** was well retained. To the contrast, the olefinic site of the free standing control substrate allylbenzene well participated in the difunctionalisation under the identical catalytic condition (Supplementary Fig. 30)[43]. It was deduced that the spatial proximity between the **TPA** moiety and the docked carbonyl of substrate might facilitate the formation of a product-like late transition state[52] by restricting the photogenerated CF₃ radical near to the fixed α,β-unsaturated olefinic group in the confined space, which might be important to kinetically distinguish the bonded and unbonded olefinic sites within the same molecules (Supplementary Fig. 31). When the biologically interesting steride scaffolds containing olefinic sites were merged together with the α,β-unsaturated esters, the corresponding steride derivatives **7p** and **7q** delivered excellent regio- and diastereocontrol simultaneously, and the carbonyl-adjacent olefinic terminals of substrates were converted with perfect diastereoselectivity while retaining single or even multiple carbonyl-free alkenes in the fragments of cholesterol and ergosterol (Fig. 8b, p and q), illustrating the potential of this heterogeneous synergistic photoredox-copper catalytic system in drug discovery.

Then, DFT calculations were performed to investigate the role of readily accessible Cu(II) sites of Cu–**Twisted** in the Hammond postulate-typed discrimination of different olefinic sites during the synergistic catalytic difunctionalisation. As shown in Fig. 9, the energy profiles of the reaction pathways of the carbonyl-adjacent olefinic site (pathway **a**, in red) of **7l** and the opposite-side carbonyl-free allyl terminal (pathway **b**, in blue) were computed at the B3LYP/6-31G(d)-LANL2DZ level and compared. Addition of the CF₃ radical to the olefinic site nearby the docked carbonyl **7l** occurs via transition state **TS I** by an energy barrier of 16.76 kcal mol⁻¹ to form carbon-centred radical intermediate **II**. Subsequently, the chloride anion coordinates to the copper centre and replaces the carbonyl moiety, and the copper–Cl species mediates Cl-atom transfer to form **8l**[53]. The overall transformation via pathway **a** is exergonic by 12.19 kcal mol⁻¹. In contrast, addition of the CF₃ radical to the carbonyl-free allyl terminal of **7l** occurs through transition state **TS I'** with a higher energy barrier of 25.6 kcal mol⁻¹, and all subsequent steps also proceed through higher energy barriers despite pathway **b** being more thermodynamically favourable overall (exergonic by 24.21 kcal mol⁻¹).

Consequently, the two thermodynamically allowed pathways were distinguished within coordination polymer, and the difunctionalisation of olefinic moiety that adjacent to carbonyl binding site in the same molecules was kinetically favoured (Fig. 8b and Supplementary Table 6). Moreover, it was revealed

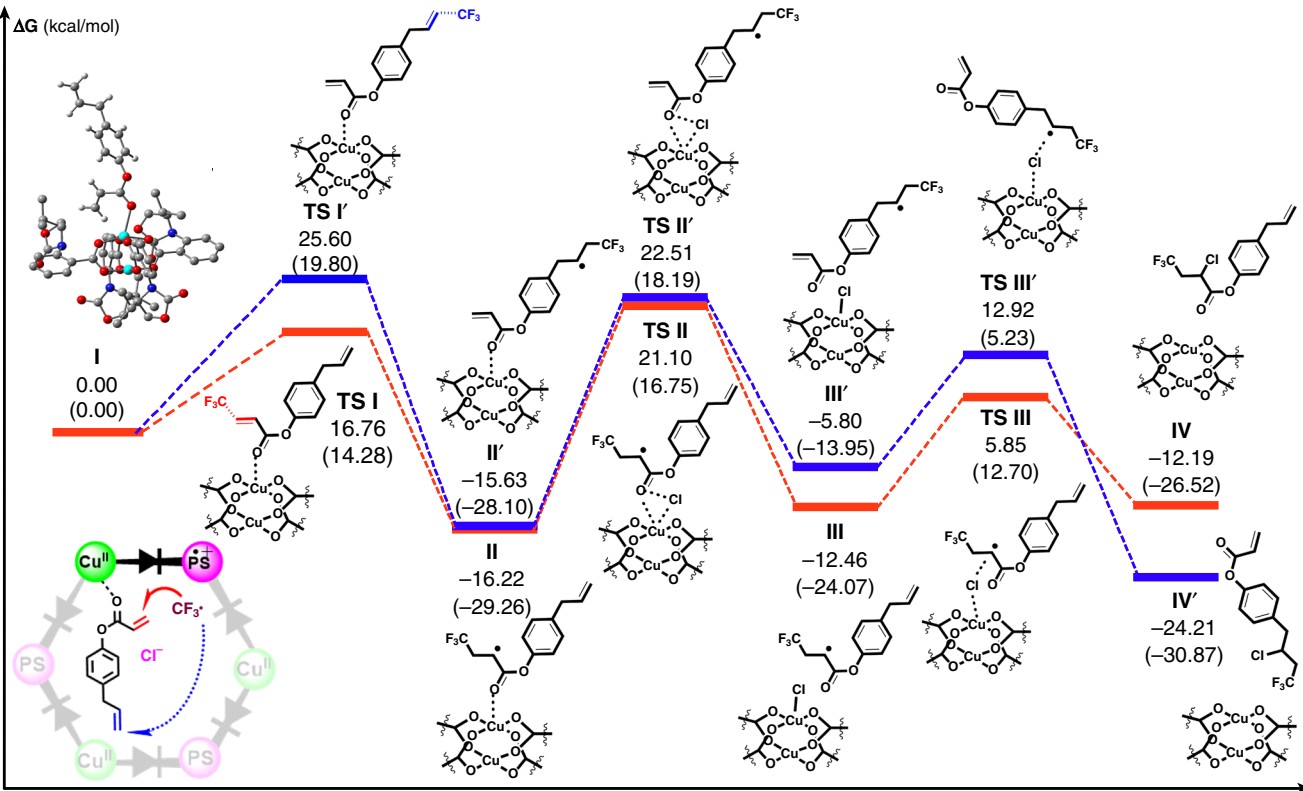

**Fig. 9 DFT study on the site selectivity.** The computed relative free energies and electronic energies (in parentheses) are given in kcal mol⁻¹. Inset, upper left (intermediate **I**): simplified DFT model of **7l** adsorbed on the Cu(II) node. Cu cyan, C grey, O red, N blue. Inset, lower left: conceptual illustration of the product-like late transition state.

that the adsorption of substrate **7l** was more favoured than that of product **8l** (the comparison of DFT calculated free energy changes, −12.44 < −11.03 kcal mol⁻¹, see Supplementary Table 7). This result implied that the generated product may be crowded out from Cu(II) centre by the competitive adsorption of a substrate molecule to trigger the new round of reaction, facilitating the recyclability of the heterogeneous catalysts.

## Discussion

In summary, we have developed a novel heterogeneous approach to combining photoredox and copper catalysis for decarboxylative C(sp³)–heteroatom couplings and site-selective trifluoromethylation difunctionalisation of olefins by using Cu (II)–dye coordination polymer. The twisted conjugation between photoreductive **TPA**-moiety and the polar carboxylate–Cu(II) node in coordination polymer featured the photoelectronic behaviour of molecular diode array for hampering the futile inter- and intramolecular fluorescence quenching of Cu(II) site towards **TPA**-based ligand, rectifying the productive unidirectional electron transfer route. Thus, the high local concentration of Cu(II) sites that retained during PET process exerted the unique modulation effects on substrates, reagents, and radical intermediates, endowing the synergistic photoredox and copper catalysis with the much broader adaptability to different types of value-added reactions, the concise and easy handling reaction conditions, and the distinctive reaction selectivities. Fabrication of the coordination polymer employs economic organic dyes in place of noble-metal-complex photosensitisers, and the heterogeneity of the reactions facilitates catalyst recovery after use, circumventing the residual heavy metal issues of usual homogeneous protocols. This package deal paves the way for designing novel synergistic photoredox and high-valent transition-metal catalytic systems from

an intrinsic perspective of molecular device, which might trigger tremendous new possibilities in both fields of pharmaceuticals and photoelectronics.

## Methods

**Materials and measurements.** All commercial chemical sources and experimental details for ¹H NMR, ¹³C NMR, ¹⁹F NMR, HRMS, IR, thermogravimetric analysis, EPR, CV, EIS, single-crystal X-ray crystallography, PXRD, and photoelectrochemical measurements are provided in the Supplementary Material.

**Synthesis of Cu−Twisted.** A mixture of $H_3L$–Twisted (0.02 mmol) and Cu(NO₃)₂ · 3H₂O (0.08 mmol) were dissolved into solvent mixture of DMF/MeOH (3 mL/1 mL), in a vial. After addition of 3 drops of HCl (3 M, aq.), the vial was sealed in a Teflon-lined stainless steel autoclave and heated at 80 °C for 3 days. The reaction system was then cooled to room temperature at a rate of 5 °C h⁻¹. Green block crystals were collected in 70% yield (based on ligand). Ememt analysis (calcd., found for $C_{198}H_{204}N_{14}O_{51}Cu_3$): H (5.43, 5.58), C (62.81, 62.76), N (5.18, 5.09); IR (KBr): 3390, 2962, 1729, 1597, 1517, 1489, 1438, 1386, 1323, 1262, 1214, 1189, 1148, 1051, 1015, 969, 859, 830, 766, 729, 681, 654, 521 cm⁻¹; ¹H NMR (400 MHz, DMSO-$d_6$/DCl): δ7.97 (d, $J = 8.2$ Hz, 3H), 7.77 (d, $J = 8.7$ Hz, 6H), 7.74 (dd, $J = 8.5$ and 1.6 Hz, 3H), 7.65 (d, $J = 1.3$ Hz, 3H), 7.24 (d, $J = 8.6$ Hz, 6H), 4.49–4.39 (m, 6H), 4.26 (dd, $J = 6.8$ and 4.8 Hz, 3H), 1.97–1.91 (m, 3H), 0.94 (d, $J = 6.8$ Hz, 9H), 0.82 (d, $J = 6.9$ Hz, 9H).

**Decarboxylative C(sp³)–N coupling of iodonium carboxylates.** To a pre-dried Pyrex tube equipped with a cooling water system was added specified amounts of catalyst (2.5 mol%, 7.5 μmol), N-nucleophile (1 equiv., 0.30 mmol), and hypervalent iodine (2 equiv., 0.60 mmol), then the tube was sealed and subjected to three vacuum/N₂ refill cycles. After adding anhydrous degassed 1,4-dioxane (6 mL) and base BTMG (2 equiv., 0.60 mmol) by syringe, the reaction mixture was stirred and irradiated with 405-nm LEDs for 5 h. The catalyst was filtered, the filtrate was concentrated under reduced pressure, and the product was isolated via flash chromatography on silica gel.

**Decarboxylative C(sp³)–heteroatom coupling of NHPI esters.** To a pre-dried Pyrex tube equipped with a cooling water system was added specified amounts of catalyst (2.5 mol%, 7.5 μmol), nucleophile (1.0 equiv., 0.30 mmol), and redox-active NHPI ester (2.0 equiv., 0.60 mmol), then the tube was sealed and subjected to three

vacuum/$N_2$ refill cycles. After adding anhydrous degassed MeCN (3 mL) and base DIPEA (*N,N*-diisopropylethylamine, 3.0 equiv., 0.90 mmol) by syringe, the reaction mixture was stirred and irradiated with 405-nm LEDs for 12 h. The catalyst was filtered, the filtrate was concentrated under reduced pressure, and the product was isolated via flash chromatography on silica gel.

**Trifluoromethylation-chloration difunctionalisation**. To a pre-dried Pyrex tube equipped with a cooling water system was added specified amounts of catalyst (2.5 mol%, 6.25 μmol) and substrate (1.0 equiv., 0.25 mmol), then the tube was sealed and subjected to three vacuum/$N_2$ refill cycles. After adding anhydrous degassed MeCN (1 mL), base 2,4,6-collidine (2.5 equiv., 0.625 mmol), and TfCl (2.5 equiv., 0.625 mmol) by syringe, the reaction mixture was stirred and irradiated with visible light by two 20-W household CFLs for 12 h. The catalyst was filtered, the filtrate was concentrated under reduced pressure, and the product was isolated via flash chromatography on silica gel.

## Data availability

The X-ray crystallographic coordinates for the structures reported in this article have been deposited at the Cambridge Crystallographic Data Centre (CCDC) under the deposition number CCDC 1870816 (Supplementary Table 1). These data can be obtained free of charge from The Cambridge Crystallographic Data Centre via www.ccdc.cam.ac.uk/data_request/cif. All other data supporting the findings of this study are available within the article and its Supplementary Information files or from the corresponding author upon request.

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

## Acknowledgements

This work was supported by the National Natural Science Foundation of China (21971031, 21890381, and U1608224).

## Author contributions

Y.S. and T.Z. contributed equally to this work. T.Z. and C.D. conceived the project, designed the experiments, and wrote the paper. Y.S. and T.Z. performed the experiments. X.-M.J. and G.X. analysed and refined the powder X-ray diffraction data. Y.S., C.H., and C.D. solved and refined the X-ray crystal structures. All authors discussed the results and commented on the paper.

## Competing interests

The authors declare no competing interests.
