## [Peer Review File · Nature Communications]

REVIEWERS' COMMENTS:

Reviewer #1 (Remarks to the Author):

The authors have adequately addressed the issues previously raised by 4 reviewers and have added some more data. This reviewer supported publication of this manuscript.

Reviewer #2 (Remarks to the Author):

Duan and coworkers report very interesting results on photocatalytic reactions catalyzed by Cu coordination polymers with photosensitizing ligands. As this manuscript and others published in the past few years, photocatalysis with coordination polymers/metal-organic frameworks present an interesting research area not will not only help the understanding of photocatalysis but also solve some intrinsic problems in homogeneous photocatalysis, such as the difficulty of running photocatalytic reactions in a batch mode and the limitation of shallow light penetration. The findings in this manuscript are quite surprising and the results will be of interest to a large readership of Nat. Commun. The authors have adequately addressed the concerns raised by previous reviewers for Nat. Catal. This reviewer recommends the publication of this manuscript in Nat. Commun.

Reviewer #3 (Remarks to the Author):

In this manuscript, Duan et al decorate bulky auxiliaries into photoreductive triphenylamine-based ligand to twist the conjugation between triphenylamine-based ligand and polar divalent Cu-carboxylate paddlewheel node in a novel MOF Cu-Twisted, which inhibits the harmful intermolecular/intramolecular photoinduced electron transfer from the reductive photosensitizer triphenylamine to the oxidative divalent Cu, but allows the intramolecular SET from the reduced monovalent Cu of paddlewheel node to the oxidized triphenylamine of the ligand. This diode mimic strategy makes photoreductive catalysis and divalent copper catalysis compatible with each other, which solves the very challenging task in this area, and is easily applicable to the valuable reactions of a broad range of C-heteroatom couplings and trifluoromethylation-chlorination difunctionalizations. Thus, this work represents the intrinsic advantages of heterogeneous catalysis by MOF (or the so called CP as described in this manuscript). we suggest the publication of this manuscript in Nature Communications after minor revisions.

1) In the pioneering homogeneous protocols of MacMillan and Hu, different combinations of photosensitizers and copper catalysts were used in C-heteroatom couplings when employing different kinds of radical precursors or heteroatom nucleophiles. How did the same heterogeneous catalyst Cu-

Twisted handle the reactions employing different kinds of radical precursors or heteroatom nucleophiles?

Responses to the Reviewers' comments of *Nat. Commun.*:**Reviewer 1's comments:**

The authors have adequately addressed the issues previously raised by 4 reviewers and have added some more data. This reviewer supported publication of this manuscript.

Responses: We highly appreciated the positive comment of the referee.

Reviewer 2's comments:

Duan and coworkers report very interesting results on photocatalytic reactions catalyzed by Cu coordination polymers with photosensitizing ligands. As this manuscript and others published in the past few years, photocatalysis with coordination polymers/metal-organic frameworks present an interesting research area not will not only help the understanding of photocatalysis but also solve some intrinsic problems in homogeneous photocatalysis, such as the difficulty of running photocatalytic reactions in a batch mode and the limitation of shallow light penetration. The findings in this manuscript are quite surprising and the results will be of interest to a large readership of *Nat. Commun.* The authors have adequately addressed the concerns raised by previous reviewers for *Nat. Catal.* This reviewer recommends the publication of this manuscript in *Nat. Commun.*

Responses: Many thanks to the referee for the positive comment.

Reviewer 3's comments:

In this manuscript, Duan et al decorate bulky auxiliaries into photoreductive triphenylamine-based ligand to twist the conjugation between triphenylamine-based ligand and polar divalent Cu-carboxylate paddlewheel node in a novel MOF Cu-Twisted, which inhibits the harmful intermolecular/intramolecular photoinduced electron transfer from the reductive photosensitizer triphenylamine to the oxidative divalent Cu, but allows the intramolecular SET from the reduced monovalent Cu of paddlewheel node to the oxidized triphenylamine of the ligand. This diode mimic strategy makes photoreductive catalysis and divalent copper catalysis compatible with each other, which solves the very challenging task in this area, and is easily applicable to the valuable reactions of a broad range of C-heteroatom couplings and trifluoromethylation-chlorination difunctionalizations. Thus, this work represents the intrinsic advantages of heterogeneous catalysis by MOF (or the so called CP as described in this manuscript). we suggest the publication of this manuscript in *Nature Communications* after minor revisions.

Comment 1: In the pioneering homogeneous protocols of MacMillan and Hu, different combinations of photosensitizers and copper catalysts were used in C-heteroatom couplings when employing different kinds of radical precursors or heteroatom nucleophiles. How did the same heterogeneous catalyst Cu-Twisted handle the reactions employing different kinds of radical precursors or heteroatom nucleophiles?

Responses: We highly appreciated the enlightening comment of the reviewer.

C(sp³)-heteroatom couplings are pharmaceutically important but challenging owing to the reliance on the synergy of generation and redox reaction of radical species, and the synergistic photoredox and copper catalysis confers new synthetic possibilities in the C(sp³)-heteroatom couplings. The major challenge limits the direct combination of photoredox and Cu(II) catalyst was the strong quenching effect of Cu(II) ions towards the highly reductive excited states of photosensitizers in solution phases.

In homogeneous pioneering protocols, photoredox and copper catalysis was compromised by exquisitely

tuning the reaction conditions to kinetically balanced off the excited-state quenching effect of *in situ* generated Cu(II) ions. Moreover, the use of different-typed substrates or radical precursors generally required remarkably varied reaction conditions (such as different combinations of photosensitizers and copper catalysts, as mentioned by the referee) and massive optimisation efforts. Those excellent and highly skilful pioneering works of MacMillan, Hu, and etc. supplied a “circuitous” strategy for synergising photoredox and copper catalysis and provided enough details of mechanistic perspectives for supporting our research.

Compared with the above-mentioned “circuitous” kinetic turning strategies, we utilised a heterogeneous approach to blocking the intermolecular quenching of Cu(II) ions towards photoreductive excited-state dye-based ligand and simultaneously tuning the intramolecular electronic communication between them. In this heterogeneous “confronting” paradigm, the diode-like unidirectional electron transfer routes compromised the direct combination of photoredox and Cu(II) catalytic steps to solve the long-termly unsettled issue of this regime; moreover, the unique coordination activation of Cu(II) sites helped to enrich and pre-organise the substrates, reagents, and intermediates within confined environments, which basically met the kinetic requirements of reactions to circumvent the needs of exquisite and tedious kinetic tuning works in homogeneous protocols, endowing the Cu-**Twisted** catalysed pharmaceutically valuable applications with much broader adaptability to different types of reactions (including but not limited to C(sp³)-heteroatom couplings, trifluoromethylation-chlorination difunctionalisations), concise and easy handling reaction conditions.

Best regards

Sincerely,

Chunying Duan, Professor,
State Key Laboratory of Fine Chemicals
Dalian University of Technology

Email: cyduan@dlut.edu.cn

Aug. 25th, 2020